# Vaccines safety and maternal knowledge for enhanced maternal immunization acceptability in rural Uganda: A qualitative study approach

Dan Kajungu[1,2] *, Michael Muhoozi[1], James Stark[3], Daniel Weibel[4], Miriam C. J. M. Sturkenboom[2]

1 Makerere University Center for Health and Population Research (MUCHAP), Kampala, Uganda, 2 Julius Global Health, University Utrecht Medical Center, Utrecht, The Netherlands, 3 Putnam Associates, Boston, Massachusetts, United States of America, 4 Weibel Consulting, Den Haag, The Netherlands

☯ These authors contributed equally to this work.
‡ These authors also contributed equally to this work.
* dan.kajungu@gmail.com

## Abstract

### Background

Maternal immunization is a successful and cost-effective public health strategy. It protects pregnant women and their infants from vaccine-preventable diseases. Uganda is exploring new vaccines for pregnant women like replacing Tetanus Toxoid (TT) with Tetanus-Diphtheria (Td). Research on knowledge, attitudes, beliefs, and willingness among pregnant women is needed before the introduction of vaccines for pregnant women. This study was aimed at exploring maternal knowledge, attitudes, willingness, and beliefs towards maternal immunization among pregnant women in rural Uganda.

### Methods

This was a qualitative descriptive study. Ten focus group discussions (FGDs) were conducted at antenatal care (ANC) clinics and in a rural community of Uganda. Five key informant interviews (KIIs) were done with health workers, for triangulation. Considering context and research characteristics, data were collected and thematically analyzed.

### Results

Women were familiar with the importance of maternal vaccines, had positive attitudes, and expressed willingness to take them. Acceptance of a new vaccine could be affected by worries of pregnant women and that of their partners, who influence health seeking decisions in a home concerning adverse events, following the maternal immunization (AEFI). There were misconceptions about introduction of vaccines such as the belief that vaccines treat malaria and general body weakness, and being used as guinea pigs to test for the vaccine before its introduction to the larger population.

**Data Availability Statement:** All relevant data are within the manuscript and its Supporting Information files.

**Funding:** This work was supported by the GCRF Networks in Vaccines Research and Development which was co-funded by the MRC and BBSRC. Grant Number: IMPRINT Network-ITCR079018. The funder provided support in the form of salaries for authors DK and MM but did not have any additional role in the study design, data collection and analysis, decision to publish, or preparation of the manuscript. The specific roles of these authors are articulated in the 'author contributions' section.

**Competing interests:** JS and DW are employed by commercial companies: Putnam Associates and Weibel Consulting respectively. They have no known competing financial interests or personal relationships that could have appeared to influence the work reported in this paper. This does not alter our adherence to PLOS ONE policies on sharing data and materials.

## Conclusion

A range of diverse sentiments and beliefs may affect uptake and acceptability of vaccines that are introduced in communities. For instance, ignoring vaccine safety concerns may impede maternal immunization acceptability, because pregnant women and their husbands are concerned about AEFI. Moreover, husbands make all health-seeking decisions at home, and their opinion is key, when considering such interventions.

## Introduction

Maternal immunization is one of the most successful and cost effective public health strategies. Successes in protection of pregnant women and their infants from vaccine-preventable diseases, especially maternal and neonatal tetanus, are directly associated with reduced morbidity and mortality [1, 2]. In fact, immunization in pregnancy is shown to reduce neonatal mortality and illnesses such as influenza [3], pertussis [4], and tetanus [5].

Maternal immunization is vaccination of women during pregnancy, to induce a protective immune response in the mother, which increases the trans-placental transfer of specific Immunoglobulin G (IgG) to the infant, for protection of the infant against specific infections [6]. The World Health Organization (WHO) and the United States of America Centers for Disease Control and prevention (CDC) recommend use of maternal tetanus vaccine, Influenza vaccine, and Tdap vaccine, to provide protection of infants through protective maternal antibodies [7–9]. Evidence exists to show the maternal immunization effectiveness [10, 11] and safety [12, 13].

In Uganda, only Tetanus Toxoid (TT) or Tetanus Diphtheria (Td) are administered to pregnant women. Knowledge, attitudes and beliefs on maternal vaccines have not been studied in Uganda. Adverse outcomes due to low maternal vaccines uptake continue to persist. In fact, an estimate of 46% of the deaths that occurred during neonatal period, a bigger proportion were largely vaccine preventable infections [14]. The Ugandan national coverage for TT second dose (TT2+) is below the 80% target, with the National coverage among pregnant women at only 49% in 2011 [15]. This can partly be attributed to limited knowledge among pregnant women, their poor attitude about immunization, their failure to attend all ANC visits, lack of training of the Village Health Teams (VHTs) on the importance of TT vaccination for pregnant mothers, limited health education to pregnant mothers, and maternal vaccine safety concerns. Studies have shown that healthcare providers [16, 17], knowledge of patient and provider [18], improved ANC attendance and surveillance systems [17, 19] play an important role in maternal vaccination uptake.

There are global efforts to improve coverage and introduce new vaccines for both neonates and pregnant women. For instance, there are efficacy and safety studies ongoing in low- and middle-income countries in respect of Pneumococcal Conjugate Vaccine and Heptavalent pneumococcal conjugate vaccine, for pregnant and lactating women. International initiatives like the Global Alliance for Vaccines and Immunizations (GAVI) are working with public and private sectors with the shared goal of creating equal access to new and underused vaccines [20]. There is also the Partnership for Influenza Vaccine Introduction, which works in concert with WHO programs, to help countries prepare for pandemic influenza [21], and to support countries' efforts to control and prevent seasonal influenza and to create sustainable, seasonal influenza vaccination programs in low and middle income countries [21].

Uganda, like other low- and middle-income countries (LMICs) is in the process of replacing TT with Td [22]. There are other novel vaccine candidates under development, however,

research about maternal vaccine knowledge, attitudes, and beliefs has been sporadically conducted in LMICs [7].

## Thematic framework

The study adopted the Andersen and Newman Behavioural Model [23] for health service utilization to permit systematic identification of factors that influence individual decisions, the environmental and need factors for vaccine acceptance. This was then modified it using Handy et al. 2017 [24] for interrelationships that drive vaccine acceptance. In order to understand vaccine acceptance, there is need to explore the underlying factors at various levels. In this case, the environmental factors were considered to be the underlying ones. These are mostly health systems related, and include availability of the needed vaccines at the service points, availability of healthcare workers and the attitude of healthcare workers towards pregnant women. The other factors affecting acceptance of maternal vaccines by pregnant women are the predisposing factors, the enabling factors, and the factors related to the need. These, however, tend to be affected by the environmental factors and are person specific as illustrated in Fig 1.

Understanding of mothers' readiness, knowledge, attitude and beliefs towards maternal vaccines, is essential to determine acceptance and uptake of vaccines, and vaccine hesitancy in maternal immunization in the African context. The aim of this study was to investigate the readiness, knowledge, attitudes and beliefs of pregnant women in Uganda towards maternal immunization.

## Methods

### Research method

The cross-sectional study was carried out among pregnant women in the community, and those attending ANC for a period of 2 months between June 2019 and July 2019. The study

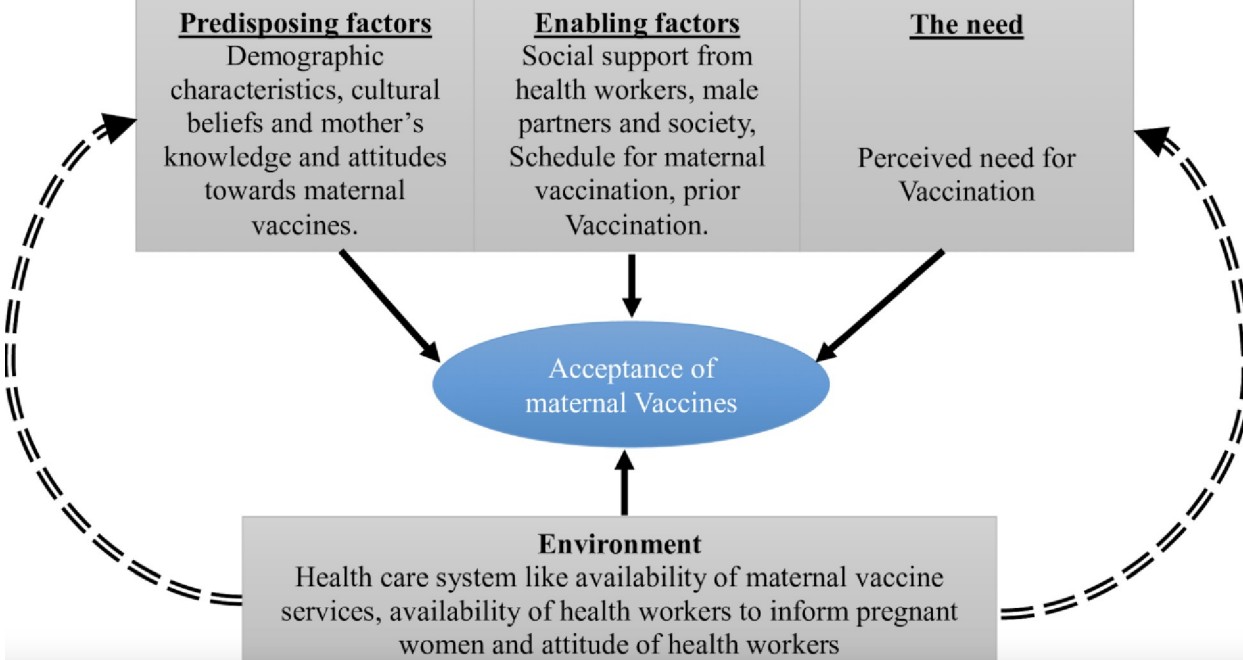

**Fig 1. Perceived usefulness of external factors that inform acceptability of new vaccines introduced for maternal immunization in rural Uganda.**

utilized a qualitative descriptive design, that helped to gain an understanding of the readiness, knowledge, attitudes and beliefs about maternal immunization in rural Uganda.

## Research population

The study included pregnant women identified at ANC and maternity points of care; and from the communities of Iganga district in Eastern Uganda. The study also included health workers from two health facilities within Iganga. Participants in the community were identified by the VHTs, through purposive sampling.

## Participant recruitment and data collection

All participants that consented to participate in the study were sampled from both the community and the health centers offering ANC to pregnant women. In the twelve sub counties, the research team used purposive sampling technique to select pregnant women and health workers for face to face KII and FGD interviews respectively. We included all pregnant women available at ANC and maternity points of care in June and July 2019, at the visited health facilities in Iganga. During the same period, pregnant women in the communities were identified and invited by the VHTs for FGDs. We excluded all pregnant women unwilling or unable to provide signed informed consent (one), and those who had any illness or complications that the investigator felt would be harmful and stopped active participation, including moderate and severe pain. A total of 10 FGDs and 5 KIIs from the health facilities and communities were purposively selected from counties of Kigulu South, Kigulu North and Iganga Municipality. These were deemed adequate since participants resounded similar themes; a suggestion that the sample size was sufficient for saturation to be achieved. FGDs and KII lasted for approximately 90 minutes and 45 minutes respectively.

## Research tool

A FGD and KII guide were developed to explore the knowledge, attitudes, willingness and beliefs about maternal immunization among pregnant women. Using WHO step by step guidelines for qualitative research [25], DK, MM, DW, JS and MCJM developed the FGD and KII, basing on theoretical categories in conceptualization. These guides were pretested and piloted in a similar population outside the study areas.

## Data management and analysis

All audio recorded interviews in *Lusoga* (a local language), were translated and transcribed into English. A social scientist working with study team and village-based scouts at the Iganga Mayuge Health and Demographic Surveillance Site (IMHDSS) [26] translated the tool from English to *Lusoga*, and transcribed the audio recording in *Lusoga* to English. A consensus was reached in meetings, where inconsistencies existed. The transcripts were proof-read before importing them in a qualitative data management software—atlas.ti Version 6.0. Data coding and analysis was conducted subsequently.

The study team worked with a social scientist based at IMHDSS; first author DK has training and experience with qualitative field work in immunization, and MM has prior training and experience with qualitative research in maternal and health services research. The team developed an initial codebook, using a sample of transcripts. The developed codebook was later applied to the entire atlas project by two coders, with any emerging codes added in the process. The query reports and code-document tables were used to establish similarities in patterns and the magnitude of categories respectively.

Thematic analysis was used to identify patterns in the data that are important or interesting for use, to address the research question. Results were presented using themes with typical quotations from the pregnant women's FGDs and supported by the evidence from KIIs.

## Research validity and reliability

To achieve a complete picture of the society and the health system, women in the villages and those attending ANC, as well as health workers responsible for maternal vaccination, were interviewed. This presented an opportunity for validating results from participants regarding the study findings, and these were similar; thus reinforcing and validating the pregnant women reports. Prior to the study, an experienced study site team was trained in qualitative data collection techniques. Study site team members ensured that participants attending ANC and those in the community were provided a conducive environment to respond objectively. Experienced research assistants and village health team members (VHTs) (who carry out IMHDSS bi-annual data collection rounds), were trained on keeping the discussions lively, gentle, and friendly with a natural conversation, personal stories, laughter and sometimes disagreements; while ensuring that the discussions are in line with a discussion guide, to ensure that all the research questions are covered.

To appreciate the collective and comprehensive interpretation of the findings, the research teams were trained for one day on using separate reflexive journals to document both verbal and non-verbal responses that could not be extracted from the interview transcript. All the interviews were recorded and transcribed. To check the translation accuracy, the FGD guide was back translated from *Lusoga* to English. Inconsistencies were resolved in routine study supervisory meetings. A pilot study was conducted among a limited sample of participants (2 FGDs for pregnant women and 2 KIIs for 2 health workers) to evaluate the questions for their cultural appropriateness to the target population. Identification of new codes and probes was based on participant responses. This helped in expanding the sample size to capture and analyse data from a diversity of backgrounds.

## Ethical considerations

Approval to conduct the research was obtained from the Mildmay Uganda Research and Ethics Committee (MUREC) (IRB Number: 0402–2019, and the study was registered by the Uganda National Council for Science and Technology (UNCST Number: HS-2669). Permission was also sought from the district authorities. Written informed consent was obtained from each participant, following a detailed explanation of the research purpose.

## Results

A total of 10 FGDs and five (5) KIIs from 15 villages of rural, and semi-urban areas were conducted. A total of 90 pregnant women agreed to voluntarily participate in the study as shown in Table 1. They were individually approached by the moderator, to solicit for socio-demographic information before joining the focus group discussions. There were four (4) FGDs for women at health centers attending ANC, and six (6) from the community in both rural and semi urban settings. The median age was 24.3 years, and they were mostly Muslims, married and 75% had ever received TT. The forty main codes identified were grouped into four themes: Knowledge about maternal vaccines and immunization, pregnant women's experience with maternal vaccination, cultures, norms & beliefs on maternal vaccination and readiness to receive new vaccines during pregnancy. Additional illustrative and summarized statements within each theme are shown in S3 File.

**Table 1. Socio-demographic characteristics of study participants (N = 90).**

| Characteristics | Category | Number (%) |
|---|---|---|
| Age | 18–24 years | 53 (59%) |
| | 25–34 years | 33 (37%) |
| | 35+ years | 4 (4%) |
| Religion | | |
| | Muslim | 48 (53%) |
| | Catholic | 16 (18%) |
| | Protestant | 16 (18%) |
| | Pentecostals | 10 (11%) |
| Marital Status | | |
| | Married | 70 (78%) |
| | Unmarried | 20 (22%) |
| Highest education level | | |
| | Primary | 43 (48%) |
| | Secondary | 41 (46%) |
| | Above secondary | 3 (3%) |
| | None | 3 (3%) |
| Ever received TT[a] | | |
| | Yes | 75 (83%) |
| | No | 15 (17%) |

[a]Tetanus Toxoid Vaccine

## Knowledge about maternal vaccines and immunization

Overall, pregnant women were knowledgeable and aware about the importance of maternal vaccination. Many were able to mention that vaccination prevents mothers and children from diseases like tetanus, measles etc. Many participants also knew that vaccination protects mothers from infection and provide immunity to the born and unborn babies. One of the participants stated that;

> "*As pregnant women. . .., we are vaccinated Tetanus because a mother may deliver along the road and this vaccination for Tetanus saves the mother. Even when you deliver at home, the umbilical cord should be handled carefully and does not get Tetanus because of the vaccine injected during pregnancy*" (FGD7).

The participants knew that they were receiving Tetanus Toxoid for protection of the baby as explained by one of the participants.

> "*Even when you give birth to a child, the baby will be free from tetanus. We hear tetanus is contracted during delivery.*" (FGD4).

Some women had an idea about Hepatitis B (Hep B) and Human Papillomavirus (HPV) Vaccine. When probed further about these two vaccines, these pregnant women confirmed that they do not receive Hep B vaccine but did not know why.

> *One of the participants said* "*as pregnant women, we do not receive Hepatitis B. . .(FGD7)* and another *participant said* "*I don't know the reasons why they don't administer Hep B to pregnant women (FDG 8)*

Some women had wrong understanding of maternal vaccination. They thought that vaccination was meant to prevent and treat malaria and general body weakness.

One pregnant woman shared that, *"What I think about vaccination is that it prevents diseases like tetanus and others like malaria, body weakness..." (FGD 4) "...it helps babies not to be attacked by malaria" (FGD 6)* one woman added.

## Pregnant women's attitudes and beliefs towards maternal vaccination

Participants generally regarded maternal immunization as very important to them and their babies, and would take up vaccines given to them. One participant noted that,

> *"When am pregnant just like I am, I must know that am supposed to go to the clinic or health centre for treatment.…..."(FGD 1).*

It was also noted that amidst their tight schedules and since maternal vaccination was only conducted on Monday and Tuesday, participants were ready to forego other responsibilities for vaccination. This respondent was quoted saying,

> *"The life of someone is very important. You can stop doing anything and attend to health workers and fulfil what he/she told you. So, I say that you have to go"* (FGD 1).

On the other hand, participants expressed concerns about adverse events following immunization (AEFI). One respondent said that,

> *"When I get tetanus vaccine, I spend about 3 days when my body is paining"* (FGD 5) and another respondent said, *"...my hands get or become paralyzed"* (FGD 5).

Participants expressed concerns about those who believed in traditional/local treatment and did not go for vaccination. One woman remarked,

> *"...because they have their traditional doctors who examine and tell them the condition of the baby in the womb and they don't see the reason why they should go to a health facility"* (FGD 3).

Other participants mentioned that some of the women preferred using traditional birth attendants and local herbs, as a remedy for complications that may occur during pregnancy. Health workers also highlighted that some of the women in the communities were using traditional remedies and would not consider going for vaccination. Participants also expressed that some pregnant women did not go for vaccination, due to their religious belief that prohibits vaccination like some Pentecostal churches. Fear of injections and HIV testing (which is compulsory for all women attending ANC) could be the reason some women fail to seek maternal vaccination.

## Experiences of pregnant women with maternal vaccination

Majority of the participants who had received maternal vaccines acknowledged that the vaccination schedule was convenient and flexible enough, to enable them utilize the vaccination services. One pregnant woman said,

> *"I think the schedule is flexible because they inform us when to go back for antenatal so you can plan very well at home putting in mind that on such a date you must come back but the problem with other women they want to be sleeping at a time when she is supposed to go to the health facility ..."* (FGD 9).

Some pregnant women had received good treatment from health workers, most especially when they went early for accessing vaccination and ANC. One participant said,

*"When you go early, there is no problem, but when you go late, they will tell you to come on the next day, since they will be tired already."* (FGD 3)

Health workers mentioned that amenities and vaccination were convenient for pregnant women, to acquire maternal vaccination. One health worker said,

*"The vaccination schedule is really convenient because it helps them to get vaccinated at the time they have come for ANC. . . ."*(KII 3).

On the other hand, there were few participants who said that the maternal vaccination schedule was not convenient, and amenities were unfavourable. One of the participants said,

*". . ..they should make it daily because there is when I went when I was sick, it was a Tuesday very early in the morning, I was even the first to arrive at the health facility and I waited up to 2:00 pm, but when the health workers came, they told me that if you are a new comer, you are supposed to come on Monday, and I went back home without even a medicine. . .."* (FGD 9).

Another participant said,

*"Pregnant women are prone to diseases like candida. Toilets at the facility are very dirty and you don't have any way of protecting yourself and you want to ease yourself."* (FGD 9).

A few participants shared the fact that they had experienced long waiting time and different instances of stock outs for drugs including vaccines. Some health workers also acknowledged the fact that there was long waiting time, which was attributed to big patient health worker ratio. One health worker said,

"*Women who come for ANC are always very many, and you have to work for long hours to complete all of them and it is tiring to. . . ."*(KII 4).

Other participants mentioned having had harsh, rude and abusive experiences with the health workers, while attending ANC. One of respondents said

"*Some health workers abuse patients telling them that they are dirty."* (FGD 1).

Another woman said that,

"*Health workers abuse them. For example, if they see you aged and you had come for vaccination, they abuse you."* (FGD 3).

## Pregnant women's willingness to receive new maternal vaccines

Women were willing to receive new maternal vaccines, as long as they acquired proper sensitization. Women were particularly interested in knowing the disease the vaccine prevents, schedule, when to receive it and its safety, to both themselves and their unborn babies. Majority of the participants also expressed concerns about the adverse effects due the vaccine that may be introduced. One of the respondents mentioned that

"*If the vaccine has no side effect to the baby and the mother, I will take it……..*" (FGD 5).

When health workers were asked what women are worried about, only one mentioned that the pregnant women and their partners were concerned about AEFI. One nurse narrated a conversation with one pregnant woman,

"*…..she told that her husband said that if it (vaccine) can cripple an old child, then how about the unborn one…*"(KII 1).

A few participants mentioned that some women believed that they were used to test for the safety of new vaccines, and thus they would not take the vaccines until its safety is confirmed by others who had received it. One of the participants said that, "*If the vaccine is introduced, I first wait for example Hepatitis B. I have to wait and receive it later, after some people had been vaccinated (laughing)*" (FGD 7).

The other respondent said that,

"*For us as people, we are prepared but the problem is that they tell us that [the President of Uganda] wants to test the drugs on us the Ugandans from this side, so it's after finding out that the drug (vaccine) has no problem, that is when it's taken to his home area for administration*" (FGD 2).

## Partner involvement in decision making for maternal vaccination

Pregnant women generally recognized the role of their husbands in decisions making and reminders regarding vaccination, even without accompanying them and their children to immunization centers. Other participants acknowledged direct engagement of partners going for ANC and maternal immunization at health centers. Some health workers also encouraged pregnant women to come with their partners for ANC visits.

However, some women argued that partners do not play an important role in the maternal immunization, as they would sometimes not provide transport or company to the health facilities. One of the participants said that,

"*When I tell my husband that am going for antenatal, he tells you to walk, yet you are tired you move without reaching you…*" (FGD9).

A few pregnant women also said that some husbands to the women in the communities always express concerns of AEFI, while referring to incidences and rumors of AEFI.

Another woman commented that,

"*Some husbands refuse us because they say vaccination cripples children.*"

Some of the women believed that the health facilities strategies to encourage pregnant women to come with their partners were not effective, since women are not comfortable with moving with husbands. Moreover, the women who did not dress well found it embarrassing to come to the health facilities. A participant said that,

"*They are dirty. One says I will not go because I will be a laughingstock, since my husband didn't buy me maternity dress and I have rags*" (FGD 4).

A small number of participants believed that this system discouraged pregnant women from accessing vaccination services. One pregnant woman said that,

*"Some fail because they want to go with their husbands. There is a system in health facilities where they first work on pregnant women accompanied by their husbands. If the husband refuses to accompany them, they also don't go"* (FGD 10).

## Discussion

Few studies assessing knowledge, attitudes and beliefs around maternal immunization have been conducted, both in an urban and rural setting as shown in the literature. This qualitative study was conducted among the Ugandan pregnant women in a semi-urban and rural settings. It highlights knowledge, attitude, beliefs and willingness of pregnant women to receive maternal vaccines. Overall pregnant women were knowledgeable about the importance of maternal immunization in providing immunity to the born and unborn babies, and had a good attitude towards maternal immunization. This finding is in agreement with other studies done in Africa [27, 28]. Messages could be well designed to focus on the impact of the disease on the infant, to increase likelihood of vaccination [29, 30].

Most women understood that they only received TT vaccine, but not Hepatitis B or Human papilloma vaccine (HPV) vaccines during their pregnancy while attending ANC. However, they do not know why they never received Hepatitis B and HPV vaccine. Lack of knowledge about other vaccines and the reasons why pregnant women are exempted from some vaccines and not the others, may prime vaccine hesitancy of any new vaccines to be introduced. This is consistent with research done in Mexico, that demonstrated that knowledge about pertussis vaccination was independently associated with the intention to receive maternal Tdap amongst pregnant women [31].

Although a small number of pregnant women viewed maternal vaccines as preventive measures for diseases like malaria and body weakness, this misinformation was also found by a recent Zambian study [32]. There is potential for vaccine hesitancy in such instances. For example, a woman who gets vaccinated with a specific maternal vaccine may hold high expectations, which include treatment of other diseases. Therefore, when she later develops such diseases like malaria that need treatment, it may affect her enthusiasm towards future vaccination, and resort to local remedies.

Pregnant women regarded maternal immunization as very important to them and their babies and are ready to go for vaccination, as long as it protected them and their babies. They however, expressed concerns regarding AEFI citing experiences regarding other vaccines other than maternal vaccines. With decreased perceived risk of vaccine-preventable diseases, fear for AEFI increases which may reduce compliance with vaccination [33]. This allows for hesitancy, compromise of coverage and re-emergence of the maternal vaccine preventable diseases [33, 34].

Despite rumors, community myths and misperceptions, pregnant women generally have trust and high expectations in the health care providers and the safety of maternal vaccines [35]. Women are emotionally attached to their pregnancies and highly consider the wellbeing of their children [36] and therefore, they need to be confident that nothing will cause any unwanted adverse outcome of their pregnancy. To mitigate worries of AEFI for pregnant women about potential risks to them and the developing fetus, there is need for a robust AEFI surveillance systems that specifically target pregnant women and their infants [37]. The passive surveillance systems could add a question about pregnancy and pregnancy outcome status to their routine AEFI surveillance reporting forms, to facilitate the process for causality assessment of serious AEFI in pregnancy and reporting of findings [37]. An approach that addresses any reactions as a result of maternal vaccines needs to be well communicated to the

beneficiaries, to alleviate concerns of AEFI, while addressing the risk/benefit determinant (epidemiological and scientific evidence) explanations, in agreement with a number of studies done [38]. The surveys of active and passive surveillance for AEFI in pregnant women and their infants need to be repeated regularly, to verify if improvement has occurred, so as to build more trust in the system for detection of AEFI [37].

In the Iganga and Mayuge districts, 73% of women have used herbal medicine at least once during their pregnancy [39], and between 20%-89.9% of women in Africa use herbal medicine during their pregnancy [40]. Some participants in this study expressed concerns about those who believed in traditional/local treatment and religious practices like in some Pentecostal churches, that prohibited maternal vaccination. Such beliefs and rumors are said to potentially affect vaccine acceptability. Therefore, there is a need for development of an approach that is culturally sensitive, to correct any fallacies, and lessen issues related to vaccine hesitancy in LMICs, as recommended by earlier studies [32, 38].

Pregnant women were willing to receive new maternal vaccines, as long as they acquired proper sensitization, and were aware about the importance of the vaccine. However, women were concerned about the vaccine related adverse events that may be introduced. Yet maternal immunization programs not only in Uganda, but also other LMICs, have no evidence on sensitization about maternal vaccination [41, 42]. This is compounded by the rare, but important misconceptions surrounding the introduction of new vaccines. Some believe that the new vaccine would be tested on them before they are given to the rest. These beliefs need to be countered by sensitization efforts, which leverage on the political efforts to help community members gain confidence in the vaccination programs, and other health care services.

There is a general agreement that the current maternal vaccination schedule was convenient and flexible enough. Flexible schedules make uptake and completion of new maternal vaccines easy, which are important in maximising the advantages associated with maternal immunization and uptake. This is in agreement with a study in Australia that showed that high risk groups benefited from an accelerated schedule and increased the likelihood of completion [43]. The issue of vaccination schedule and amenities are health system issues, and were attributed to the set time by the facilities for attendance to ANC. Strategies that improve ANC attendance need to integrate messages for encouraging maternal vaccination among the poorest, single, multiparous women, and those who do not deliver at health facilities. Different studies have associated ANC attendance with timely vaccination, and adherence to the vaccination schedule [44, 45].

Behavior of a few of the health workers and how they treat pregnant women was termed as harsh, rude and abusive, which is a massive inconvenience while attending ANC. This is similar to findings in Ethiopia and Tanzania [46, 47]. Evidence has shown that a recommendation from a health care professional for vaccination is the most important factor in decision-making for uptake of a maternal vaccine [16]. This can positively or negatively influence how women, their partners and families perceive and experience maternal health care, and ultimately have an effect on ANC attendance [35, 48]. The health system needs to develop and maintain trust of pregnant women, through building capacity of health workers to uphold ethical values and make favorable health education and literacy activities in both routine and outreach programs.

Personal hygiene and dress code are some of the concerns that could stop women from taking up maternal immunisation. Some pregnant women feel that health workers mishandle them if they are dirty, while others are not able to come with their husbands to the clinic because they are not decent or even dirty. Studies in an African context have shown that women delay to attend ANC waiting for new clothes [29, 49]. Health promotion and education

campaigns targeting maternal health services need to integrate communications that promote self-esteem, personal hygiene and non-discriminative practices for ANC attendance.

This study highlighted the importance of partner support in the maternal immunization initiatives. This is in agreement with a study done in Uganda that demonstrated that male involvement in immunization of children was important, for decision making of women to vaccinate their children [27]. However, contrary to their findings; some of the pregnant women felt that visiting the health facility with the partners for ANC and maternal immunization could delay decision making, cause embarrassment; which could potentially frustrate vaccine acceptance. Studies have shown that norms relating to fathers' participating in ANC contribute to men feeling shy, embarrassed or ashamed to attend ANC with their pregnant partner [50, 51]. Polygamy is acceptable and practiced in this community, largely because Islam is the dominant religion. Consequently, men in polygamous relationships are less likely to accompany their partner to ANC, for fear of being perceived as favouring one wife over others, which may cause conflict in the household [50, 51]. No wonder many women felt that it was their responsibility to get vaccinated for their good and that of their children. There were suggestions that some women could not come for vaccination due to failure of their husbands to accompany them or provide transport to the health facilities. While feasibility of home vaccination for pregnant women has not been determined, use of VHT and mobile phone consultations and mobile clinics interventions for hard to reach women has been recommended [52, 53].

The partner's worries about AEFI, was an important concern for women, which could affect the decision making around maternal immunization and acceptability. Integration of health education messages, that cover vaccines and their safety targeting partners' role and involvement in maternal immunization, is important in facilitating vaccine acceptance and uptake. These factors must be considered when introducing a new vaccine, as they have the potential to enable or obstruct vaccine uptake.

## Limitations

This study was done in rural area with some semi urban settings in Uganda and results of this study need to be taken in the context in which the study was done. Education and health literacy in rural areas is usually low [54], but very vital for maternal vaccination [55]. Future studies need to explore maternal vaccine willingness, knowledge, attitudes and beliefs, and impact of education and health literacy on maternal vaccination in an urban setting, where it is expected to have differing levels of education and social economic status.

## Conclusion

Maternal vaccines have great potential to reduce the global burden of infant morbidity and mortality; with their unique position to access the infant's immune system, through maternal antibodies, before a child vaccine could be effective. However, the existence of safe and effective maternal vaccines will only be useful, if mothers and their partners decide to use them with less or no worries about the AEFI.

Maternal immunisation knowledge, attitude and willingness for pregnant women, was generally positive. However, factors such as religion, cultures, fear for AEFI for new maternal vaccines, and wrongly believing that vaccines cure diseases like malaria and reduce general body weakness, could counteract maternal vaccination acceptability.

## Recommendations

Health workers need to be trained using consistent message and provide proper training and orientation providers, on the importance of maintaining respectful and compassionate care at all times.

Efforts to improve education levels and health literacy of the community members and health care workers to some extent needs to be prioritized. Programming for maternal vaccination needs to address the individual or societal concerns, values, beliefs and norms of pregnant women, considering the context in which they are. This can be done by improving and establishing community engagements, sensitizations, outreaches and education programs on maternal vaccinations, to help reduce the burden of diseases that are maternal vaccine preventable in the Ugandan context. These will help to correct misconceptions, by replacing the existing myth with new information, rather than solely debunking an individual's current belief [56].

Establishing a robust system for monitoring vaccine deployment, emphasis on safety monitoring, following immunization and disease surveillance, is also key for maternal vaccine uptake. A strengthened system for post-marketing surveillance of vaccines that enables timely AEFI reporting, will lead public confidence in vaccines, especially the new one. This can be done by intensifying efforts for active (in sentinels) and passive surveillance systems, to provide reliable data for regulators and public health authorities. These efforts need to harness collection, reporting, and comparison of data on vaccine safety in pregnancy, and contribute to the harmonization of vaccine pharmacovigilance.

Further research will be needed to explore misconceptions about vaccines, and anxieties about testing new vaccines to a small population, before they are rolled out to the rest, that could promote or discourage vaccine acceptance.

## Supporting information

**S1 File. Focus group discussion guide.**
(PDF)

**S2 File. Key informant interview guide.**
(PDF)

**S3 File. Thematic classification of additional quotes.**
(PDF)

**S4 File. COREQ checlist.**
(PDF)

**S5 File. Analysis codebook.**
(PDF)

## Acknowledgments

The authors would like to thank the Makerere University Centre for Health and Population Research (MUCHAP), the Iganga Mayuge Health and Demographic Surveillance Site (IMHDSS), for their partnership and collaboration for this work. This research would not have been possible without the support of Iganga District health office, local leaders, and community members that participated in the study.

## Author Contributions

**Conceptualization:** Dan Kajungu, James Stark, Daniel Weibel, Miriam C. J. M. Sturkenboom.

**Data curation:** Dan Kajungu, Michael Muhoozi.

**Formal analysis:** Dan Kajungu, Michael Muhoozi.

**Funding acquisition:** Dan Kajungu.

**Investigation:** Dan Kajungu.

**Methodology:** Dan Kajungu, James Stark, Daniel Weibel, Miriam C. J. M. Sturkenboom.

**Project administration:** Dan Kajungu, Michael Muhoozi.

**Resources:** Dan Kajungu.

**Software:** Dan Kajungu.

**Supervision:** Dan Kajungu, Miriam C. J. M. Sturkenboom.

**Validation:** Dan Kajungu, James Stark, Daniel Weibel, Miriam C. J. M. Sturkenboom.

**Visualization:** Dan Kajungu.

**Writing – original draft:** Dan Kajungu, Michael Muhoozi, Daniel Weibel.

**Writing – review & editing:** Dan Kajungu, Michael Muhoozi, James Stark, Daniel Weibel, Miriam C. J. M. Sturkenboom.

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
