## [Decision Letter · Decision Letter 0]

21 Aug 2020

PONE-D-20-15557

Vaccines safety and maternal knowledge are determinants of maternal immunization acceptability in rural Uganda – A Qualitative Study Approach.

PLOS ONE

Dear Dr. Kajungu,

Thank you for submitting your manuscript to PLOS ONE. After careful consideration, we feel that it has merit but does not fully meet PLOS ONE’s publication criteria as it currently stands. Therefore, we invite you to submit a revised version of the manuscript that addresses the points raised during the review process.

We look forward to receiving your revised manuscript.

Kind regards,

Holly Seale

Academic Editor

PLOS ONE

Journal Requirements:

2. Please include a copy of the interview guide used in the study, in both the original language and English, as Supporting Information, or include a citation if it has been published previously.

3. Thank you for stating the following beneath the Acknowledgments Section of your manuscript:

'Funding

This work was supported by the GCRF Networks in Vaccines Research and Development which was co-funded by the MRC and BBSRC. Grant Number: IMPRINT Network-ITCR079018.  The authors declare that they have no known competing financial interests or personal relationships that could have appeared to influence the work reported in this paper.'

'The funders had no role in study design, data collection and analysis, decision to publish, or preparation of the manuscript.'

'The authors have declared that no competing interests exist.'

We note that one or more of the authors are employed by commercial companies: Putnam Associates and Weibel Consulting.

5. Please include captions for your Supporting Information files at the end of your manuscript, and update any in-text citations to match accordingly. Please see our Supporting Information guidelines for more information: http://journals.plos.org/plosone/s/supporting-information

Reviewers' comments:

Reviewer's Responses to Questions

**Comments to the Author**

1. Is the manuscript technically sound, and do the data support the conclusions?

Reviewer #1: Yes

Reviewer #2: Yes

2. Has the statistical analysis been performed appropriately and rigorously? 

Reviewer #1: Yes

Reviewer #2: Yes

3. Have the authors made all data underlying the findings in their manuscript fully available?

Reviewer #1: Yes

Reviewer #2: Yes

4. Is the manuscript presented in an intelligible fashion and written in standard English?

Reviewer #1: Yes

Reviewer #2: Yes

5. Review Comments to the Author

Reviewer #1: This novel study is an in-depth exploration of the attitudes and knowledge regarding maternal vaccination held by pregnant women and health workers in Uganda. Though many women were aware of the importance of maternal vaccination, there were many misperceptions. Both pregnant women and health workers reported that many women use traditional medicine as a means of prevention, rather than vaccination. A barrier for 12% of the women were that they belonged to Pentecostal churches and thus are forbidden from vaccination. Participants mentioned that there are only 2 days per week that they can visit a health clinic for vaccination; some said that the clinics were dirty, but more disturbingly, many women experienced abusive or lazy behaviour from health care workers. The authors reported that many of the women had safety concerns about vaccination, as did their husband/s, particularly regarding new vaccines. Many of the participants spoke about their husbands’ involvement in vaccination: some were supportive, but some said their husbands were embarrassed by their wife’s appearance, and some did not accompany them to the clinic. The results need to be published as a means to working towards improving pregnant women’s knowledge on vaccination and vaccine-preventable diseases, improving rural Ugandan women’s access to safe health services, and subsequently increasing maternal vaccine uptake. Thank you for the opportunity to read and review this important paper.

MAJOR COMMENTS:

1. Introduction

a. Given many of the women had safety concerns, and the authors suggest utilising a vaccine safety system, vaccine safety should be discussed in the introduction. Some suggested literature (though only on influenza):

i. Foo DY, Sarna M, Pereira G, Moore HC, Fell DB, Regan AK. Early childhood health outcomes following in utero exposure to influenza vaccines: a systematic review. Pediatrics. 2020 Aug 1;146(2).

ii. Giles ML, Krishnaswamy S, Macartney K, Cheng A. The safety of inactivated influenza vaccines in pregnancy for birth outcomes: a systematic review. Human vaccines & immunotherapeutics. 2019 Mar 4;15(3):687-99.

b. A brief comment on what’s known generally on the facilitators and barriers of maternal vaccination is warranted. Suggest reading and citing the following:

i. Kilich E, Dada S, Francis MR, Tazare J, Chico RM, Paterson P, Larson HJ. Factors that influence vaccination decision-making among pregnant women: A systematic review and meta-analysis. Plos one. 2020 Jul 9;15(7):e0234827.

c. Thematic framework and figure 1:

i. I like this figure. However, it is not clear to me where the ideas have come from? Did the authors refer to other models/theories, such as the Social Ecological Model?

ii. A description on why environmental factors were considered to be the underlying factors to vaccine acceptance is needed.

2. Methods

a. Participant recruitment and data collection:

i. When did the authors stop recruiting? Did they aim to get 10 FDGs and 5 KIIs from 5 villages and 90 women? Or is this when they felt they had reached ‘data saturation’?

b. Research tool

i. an explanation of who developed it and how it was developed is warranted

c. Data management and analysis:

i. Who translated and transcribed the audio recordings from Lusoga to English? And who translated them back from English to Lusoga? Was there consistent

ii. The authors say DK and MM developed the initial codebook. What is their experience with qualitative analysis? I note that their education is provided on page 1, but it is worthwhile positioning their level of expertise here too.

d. Research validity and reliability:

i. How did the study site team members provide a conducive environment?

ii. How trained the researchers on how to document verbal and non-verbal responses?

3. Results

a. How was the information for Table 1 collected? Was it self-report on a form? Or were participants asked in the focus group interviews?

b. There are a lot of brilliant quotes. However, having so many disrupts the flow for the reader. As many of the quotes are already summarised in the sentence before it, and therefore the quotes don’t add anything extra to that point, I suggest removing the quotes on the following lines:

i. Line 281

ii. Lines 287-288

iii. Lines 293-294

iv. Line 300

v. Lines 327-328

vi. Lines 360-362

vii. Line 393 (and instead add the word “reminders” to the sentence “…husbands in decision making and reminders regarding vaccination...”).

viii. Line 398

ix. Lines 403-404

c. Similarly, to help the flow of reading for the reader, I think the results in section 3.3 should be reformatted, to include all the positive comments or comments about convenience at the beginning, followed by all of the negative comments or comments about inconvenience at the end. It is currently mixed.

d. Lines 416-421. This is a striking result. I think the authors could make this section clearer, however. Furthermore, I understand that polygamy is legal in Uganda; when the participant on line 420 says “One says I will not go….” is this one of her husbands?

4. Discussion

a. There are a lot of bold statements in the discussion, that haven’t been referenced. Please either reword or find references. Some examples of unreferenced bold statements:

i. Lines 448-449: “For example, a woman who gets vaccinated and later develops a disease like malaria may fail to seek for future vaccination.”

ii. Lines 453-454: “With decreased perceived risk of immune-preventable diseases, fear for AEFI increases which may reduce compliance with vaccination.”

iii. Lines 457 – 459: “Pregnant women generally have trust and high expectations in the safety of maternal vaccines. Women are emotionally attached to their pregnancies and highly consider the wellbeing of their child….”

b. I’d be interested in reading about the abusive behaviour some health care workers have towards pregnant women. Do other LMIC countries experience this? Given a health care providers recommendation is one of the most important facilitators of vaccination, what’s the impact of the abusive behaviour on uptake? It would also be worthwhile noting that while health care workers should receive training on using consistent messaging (Lines 551-552), some appear to need training their behaviour and bedside manners.

c. I’d also be interested in reading about the impact of husbands on maternal vaccination. For example, the shaming husbands do of their wives’ appearances and clothes. Has this been documented anywhere else in the literature? Given husbands were also important in decision-making, how does this work in polygamous relationships? Finally, given that some women weren’t able to access health clinics as their husband would not accompany them, is home vaccination an achievable, acceptable and safe option for women in Uganda? Are there any other health services delivered in the home?

d. To me, there was also a theme about cleanliness/dirtiness. Some health workers abused patients if they were dirty, some women reported on dirty toilets at clinics, and some women said their husbands wouldn’t go with them to the clinic as the women were dirty. Further discussion about this, with potential solutions if there are any, would be interesting.

e. Safety surveillance: whilst this important, I’m not sure whether this is the first strategy that should be implemented. It’s also costly and will take time to implement. Consider other more-immediate interventions that may help these women. Some guidance in these papers:

i. Krishnaswamy S, Lambach P, Giles ML. Key considerations for successful implementation of maternal immunization programs in low and middle income countries. Human vaccines & immunotherapeutics. 2019 Apr 3;15(4):942-50.

ii. Ellingson MK, Dudley MZ, Limaye RJ, Salmon DA, O’Leary ST, Omer SB. Enhancing uptake of influenza maternal vaccine. Expert review of vaccines. 2019 Feb 1;18(2):191-204.

MINOR COMMENTS:

5. Title

a. The title should be revised. The word “determinants” is usually reserved for studies that have utilised quantitative methods.

6. Article summary

a. Line 56: Is this about women in Uganda? And about maternal immunisation?

7. Key words

a. Suggest using more specific MeSH terms, such as “Health Knowledge, Attitudes, Practice” and a term to describe Uganda

8. Introduction

a. Some of the sentences are quite long (e.g. lines 69-72). Consider shortening

b. It’s important that the references used directly relate to the topic. For example, line 71 discusses tetanus, but reference #2 is not about tetanus\\

c. Line 86: How was 40% update for TT2+ estimated? What data was collected/used? Is it a reliable estimate?

d. I suggest using systematic reviews as much as possible in the introduction. For example, the following systematic review should be referenced:

i. Nunes MC, Madhi SA. Influenza vaccination during pregnancy for prevention of influenza confirmed illness in the infants: a systematic review and meta-analysis. Human vaccines & immunotherapeutics. 2018 Mar 4;14(3):758-66.

9. Methods

a. When was the study carried out and how long did it go for? It would be worthwhile adding the year/s and duration into the ‘research method’ section

b. Participant recruitment and data collection:

i. As the reader, it would be clearer to move lines 157-162 to line 153, after “interviews at both the community and health facility.” This will help the reader understand how the research team purposively selected participants

10. Results

a. Table 1:

i. Given the numbers are <100, it is advisable to round the % to the nearest whole number

ii. The total number in religion does not equal 90

iii. A space is needed between the numbers and brackets

iv. Please spell out what TT means (underneath the table)

b. Knowledge about maternal vaccines and immunisation:

i. Please clarify whether this sentence “Many were able to mention that vaccination prevents mothers and children from diseases like tetanus, measles, etc” is referring only to vaccination during pregnancy, or vaccination of people in general?

c. Lines 357-358. I think the authors could expand on this and define what they mean by “sensitization.”

d. Line 383: I suggest removing the Ugandan President’s name, and just refer to them as the President of Uganda. I.e., you would write “we are prepared but the problem is that they tell us that [The President of Uganda] wants to test the drugs on us….”

e. Lines 409-410: This quote doesn’t relate to the sentence above it. Consider using another quote.

f. Line 421: Should “rugs” be “rags”?

11. Discussion

a. Lines 439-440: distinction is needed as to whether participants are talking about vaccination prior to pregnancy, or whether pregnant women think they haven’t yet been vaccinated against hepatitis B or HPV in their pregnancy?

b. Line 442: suggest using the word “prime” here instead of “encourage”

c. Line 453: please reword “immune” to “vaccine.”

Reviewer #2: Review of manuscript for PLOS ONE

Summary: This is a well done manuscript. The title, background, findings, and discussion are very well laid out. Methods need some clarity for reproducibility. Just a few suggestions to make it a little more crisp:

Comments on

Title: No comment

Abstract:

Lines 33 &34: You need to recast this sentence. “Women expressed willingness to take new vaccines, positive attitudes towards maternal immunization, and were familiar with its importance”. Recast the sentence to follow the order in your objectives namely knowledge, attitudes and willingness. For example: “Women were familiar with the importance of maternal vaccines, have positive attitudes and expressed willingness to take them”.

Line 35: “……... affected by adverse events”, ie changing the effects to events. Also recast the statement to be more sharp and crisp. Such as “………. affected by worries of pregnant women and that of their partners’ who influence health seeking decisions in a home concerning adverse events following the martenal immunization.”

Line 36-38: “Misconceptions about introduction of vaccines like thinking that vaccines will be tested on only them before introduction in larger population, and that vaccines treat illnesses like malaria and general body weakness.” Consider this phrase “ …… of vaccines such as the belief that vaccines treat malaria and general body weakness and that of being used as guinea-pig to test for the vaccine before its introduction to the larger population.”

Background:

Line 77: It is World Health Organization and not “world ….”

Line 80: “………. are administered to pregnant women.”

Line 82: Do you mean that poor neonatal health outcomes are due to lack of studies on knowledge ………. Make it clearer.

Line 87: Be consistent. “ Immunisation” is British English. You have been using American English. In addition, there is a little flaw in the content and mechanical accuracy in that sentence. You could consider “This can partly be attributed to limited knowledge among pregnant women, their poor attitude about immunization, their failure to attend all ANC visits, lack of training of the Village Health Teams (VHTs) on the importance of TT vaccination for pregnant mothers and limited health education to pregnant mothers.”

Line 92: “ ………. middle-income countries in respect of Pneumococcal Conjugate Vaccine ……….. “

Line 95: GAVI means Global Alliance for Vaccines and Immunizations.

Methods: This section need some work.

Line 144: You had included health workers in the research population included. You failed to include them now.

Line 153: Which of the population had FGD and which one had KII. You need to be coherent and orderly.

Line 157: Readers may find it difficult to understand what you mean. I think you may look at it again and recast it to bring out your points clearly.

Line 165: Order and sequence is important as mentioned in previous comment.

Line 174: Pointless mentioning names of those that played a role here.

Results:

Line 194: “ (2 FGDs pregnant women and 2 health workers) could read “(2 FGDs for pregnant women and 2 KIIs for 2 health workers)”, or what exactly, do you have in mind.

Line 208: Change the FDG to FGD. Where exactly is the study area; rural or urban or rural and semi-urban? Be consistent.

Line 239: You use either “reason” or “why”.

Line 250: I could have preferred “….one woman added” than the word chipped in. This is scientific writing.

Line 264: AEFI means adverse events following immunization. It refers to any untoward medical occurrence which follows immunization and which does not necessarily have a causal relationship with the usage of the vaccine.

Line 283: could read “ …… highlighted that some of ……”

Line 290: “Participants also expressed that some pregnant women did not go for vaccination due to their prohibitive religious practice like some Pentecostal churches”, could read “Participants …….. due to their religious belief that prohibited vaccination”.

Line 296: This statement “Some mentioned the fear for injections and HIV testing (which is compulsory for all women attending ANC) would be the reason why some women do not seek maternal vaccination” could read “Fear of injections and HIV testing could be the reason some women fail to seek maternal vaccination.”

Line 369: “When health workers were asked what women are worried about, only one mentioned AE worries from the pregnant women and their partners” could read “When health workers were asked what women are worried about, only one mentioned that the pregnant women and their partners were concerned about AEFI.”

Line 375: The statement could be rephrased thus “Some participants believe that they were used to test for the safety of new vaccines and thus they would not take the vaccines until its safety is confirmed by others who had received it.

Line 389: Could be rephrased as “Pregnant women …………. vaccination even without accompanying them and their children to immunization centers.”

Line 400: You may want to rephrase thus “Health workers cited that they encouraged pregnant women to come with their partners for ANC visits.”

Line 417: “…………. not effective since women are not comfortable with moving with husbands and more so their husbands don’t dress well.”

Discussion:

Line 434: be consistent with the study area.

Line 452: change adverse effect to adverse event

Line 475: Rephrase as “…………... and religious practices found in some Pentecostal churches, that prohibited maternal vaccination.

Line 482: Rephrase the statement for clarity

Line 486: Rephrase the statement. It could read “Some believe that the new vaccines …………...”

Line 502: delete treatment from the sentence.

Line 504: You can rephrase as “The attitudes and behaviors of health care providers towards pregnant women is an …….”

Line 517: Consider rephrasing thus, “……… get vaccinated for their good and that of their children”.

Line 518: Consider this rephrasing “The partner’s worries about AEFI was an important concern for women ………….”

Line 537: See the comment on line 34

Line 538: Define ADR. Do you mean Adverse Drug Reaction? Since we are talking of immunization why not use AEFI instead.

6. PLOS authors have the option to publish the peer review history of their article (what does this mean?). If published, this will include your full peer review and any attached files.

Reviewer #1: No

Reviewer #2: **Yes: **Dr Uchechukwu Joel Okenwa

---

## [Author Response · Author response to Decision Letter 0]

9 Oct 2020

Reviewer #1 Point by Point Response to Review of manuscript for PLOS ONE

 1. Introduction

a. Given many of the women had safety concerns, and the authors suggest utilizing a vaccine safety system, vaccine safety should be discussed in the introduction. Some suggested literature (though only on influenza):

i. Foo DY, Sarna M, Pereira G, Moore HC, Fell DB, Regan AK. Early childhood health outcomes following in utero exposure to influenza vaccines: a systematic review. Pediatrics. 2020 Aug 1;146(2).

ii. Giles ML, Krishnaswamy S, Macartney K, Cheng A. The safety of inactivated influenza vaccines in pregnancy for birth outcomes: a systematic review. Human vaccines & immunotherapeutic. 2019 Mar 4;15(3):687-99.

Response: Thank you very much for pointing this out. A statement to illustrate evidence on maternal vaccines safety and effectiveness using the recommended and other literature has been included in the introduction. This has been included in the Introduction section line 88.

b. A brief comment on what’s known generally on the facilitators and barriers of maternal vaccination is warranted. Suggest reading and citing the following:

i. Kilich E, Dada S, Francis MR, Tazare J, Chico RM, Paterson P, Larson HJ. Factors that influence vaccination decision-making among pregnant women: A systematic review and meta-analysis. Plos one. 2020 Jul 9;15(7):e0234827.

Response: Thank you very much. A brief comment that includes facilitators for maternal vaccination has been included as per recommendation for the different evidence syntheses. The barriers have also been highlighted. These can be seen in the introduction section on line 100-102. 

c. Thematic framework and figure 1:

i. I like this figure. However, it is not clear to me where the ideas have come from? Did the authors refer to other models/theories, such as the Social Ecological Model?

Response: Thank you for highlighting this, authors used Andersen and Newman Behavioural Model and modified using Handy et al 2017 and justification for using both was provided. This can be seen under the thematic framework section on line 124-127.

ii. A description on why environmental factors were considered to be the underlying factors to vaccine acceptance is needed. Resounded

Response: Thank you, this was briefly described in lines 127-134.

2. Methods

a. Participant recruitment and data collection:

i. When did the authors stop recruiting? Did they aim to get 10 FDGs and 5 KIIs from 5 villages and 90 women? Or is this when they felt they had reached ‘data saturation’?

Response: Thank you. More clarification on how data saturation was reached has been included to illustrate the basis for which recruitment was stopped. This can be seen in methods section, sub-section participant recruitment and data collection, line 185-186.

b. Research tool

i. an explanation of who developed it and how it was developed is warranted

Response: Thank you for this notification. An explanation was included illustrate how the tools were developed using the WHO step by step guide for qualitative research basing on the conceptualization framework. This can be seen in methods section, line 191-193. 

c. Data management and analysis:

i. Who translated and transcribed the audio recordings from Lusoga to English? And who translated them back from English to Lusoga? Was there consistent

Response: A statement to clarify this has been added in line 198-201. ‘…... A social scientist working with the study team and village-based scouts at the Iganga Mayuge Health and Demographic Surveillance Site (IMHDSS) translated the tool from English to Lusoga and transcribed the audio recording in Lusoga to English. A consensus was reached in meetings where inconsistencies existed.’

ii. The authors say DK and MM developed the initial codebook. What is their experience with qualitative analysis? I note that their education is provided on page 1, but it is worthwhile positioning their level of expertise here too.

Response: Level of expertise has been added and explained in line 204 – 206. 

d. Research validity and reliability:

i. How did the study site team members provide a conducive environment?

Response: Thank you. Clarification was provided with this statement in line 222-227. ‘Experienced research assistants and village health team members (VHTs) (who carry out IMHDSS bi-annual data collection rounds) were trained on keeping the discussions lively, gentle, and friendly with a natural conversation, with personal stories, laughter and sometimes disagreements; while ensuring that the discussions are in line with a discussion guide to ensure that all the research questions are covered.’

ii. How trained the researchers on how to document verbal and non-verbal responses?

Response: Thank you, we have clarified this by showing that “a one-day training” (line 230) for the research teams was provided on how to maintain journals for the verbal and non-verbal responses to complement their experience at the study site.

3. Results

a. How was the information for Table 1 collected? Was it self-report on a form? Or were participants asked in the focus group interviews?

Response: A statement to reflect how the socio-demographic information was collected has been included (line 250-251) 

b. There are a lot of brilliant quotes. However, having so many disrupts the flow for the reader. As many of the quotes are already summarized in the sentence before it, and therefore the quotes don’t add anything extra to that point, I suggest removing the quotes on the following lines:

i. Line 281

ii. Lines 287-288

iii. Lines 293-294

iv. Line 300

v. Lines 327-328

vi. Lines 360-362

vii. Line 393 (and instead add the word “reminders” to the sentence “…husbands in decision making and reminders regarding vaccination...”).

viii. Line 398

ix. Lines 403-404

Response: Thank you for this highlight. The highlighted lines have been reformatted and some deleted to ensure the flow of results is not disrupted.

c. Similarly, to help the flow of reading for the reader, I think the results in section 3.3 should be reformatted, to include all the positive comments or comments about convenience at the beginning, followed by all of the negative comments or comments about inconvenience at the end. It is currently mixed.

Response: Thank you for the guidance. This section has also been reformatted to begin with positive comment and then followed by negative comments while ensuring coherence. This can be seen in results section line 349-406 

d. Lines 416-421. This is a striking result. I think the authors could make this section clearer, however. Furthermore, I understand that polygamy is legal in Uganda; when the participant on line 420 says “One says I will not go….” is this one of her husbands?

Response: Thank you. This was a statement about some husbands but not ‘one of her husbands’. Concerns were with the dress code for both pregnant women and their husbands. This is largely a Muslim community (53.3%) and polygamy is acceptable. 

4. Discussion

a. There are a lot of bold statements in the discussion, that haven’t been referenced. Please either reword or find references. Some examples of unreferenced bold statements:

i. Lines 448-449: “For example, a woman who gets vaccinated and later develops a disease like malaria may fail to seek for future vaccination.”

Response: Thank you for this comment. This has been altered to reflect a scenario and read thus; “…..a woman who gets vaccinated with a specific maternal vaccine may hold high expectations which include treatment of other diseases and when she later develops diseases like malaria that needs treatment, she may fail to seek for future vaccination and resort to local remedies.” This is in the discussion section line 521-524.

ii. Lines 453-454: “With decreased perceived risk of immune-preventable diseases, fear for AEFI increases which may reduce compliance with vaccination.”

Response: Reference to this assertion has been provided. This is in the discussion section line 529.

iii. Lines 457 – 459: “Pregnant women generally have trust and high expectations in the safety of maternal vaccines. Women are emotionally attached to their pregnancies and highly consider the wellbeing of their child….”

Response: Thank you for this. Statements have been rephrased and referenced using literature available as recommended. This is in the discussion section line 532-534.

b. I’d be interested in reading about the abusive behaviour some health care workers have towards pregnant women. Do other LMIC countries experience this? Given a health care providers recommendation is one of the most important facilitators of vaccination, what’s the impact of the abusive behaviour on uptake? It would also be worthwhile noting that while health care workers should receive training on using consistent messaging (Lines 551-552), some appear to need training their behaviour and bedside manners.

Response: Thank you for this helpful observation. We added something about on the behaviour of health workers towards pregnant women in the discussion. We have also referenced a study in LMICs where this was experienced. Impact of healthcare behaviour on uptake (earlier suggested) gave good insights on this and it has been used. The recommendation of training on consistent messaging has been beefed up with recommendation -very helpful observation. This is in the discussion section lines 581-588 and lines 590-596.

c. I’d also be interested in reading about the impact of husbands on maternal vaccination. For example, the shaming husbands do of their wives’ appearances and clothes. Has this been documented anywhere else in the literature? Given husbands were also important in decision-making, how does this work in polygamous relationships? Finally, given that some women weren’t able to access health clinics as their husband would not accompany them, is home vaccination an achievable, acceptable and safe option for women in Uganda? Are there any other health services delivered in the home?

Response: More details about dress code and husbands’ failure to come for ANC because of shame, embarrassment and shyness have been discussed with supporting literature from an African Context. Given that majority of the participants in the study as well as Iganga district are Muslims- polygamy was a very important fact that was recommended. A statement showing how polygamous men may fail to attend ANC for fear of being perceived as favoring one wife over others, that is also conflict in the household was highlighted. A statement acknowledging that home vaccinations feasibility for pregnant women has been included. Other options like using VHT, mobile clinics and consultations have been suggested for application. This is in the discussion section lines 604-610 and lines 613-616.

d. To me, there was also a theme about cleanliness/dirtiness. Some health workers abused patients if they were dirty, some women reported on dirty toilets at clinics, and some women said their husbands wouldn’t go with them to the clinic as the women were dirty. Further discussion about this, with potential solutions if there are any, would be interesting.

Response: Thank you. Cleanliness/dirtiness could have been a theme, authors thought that since it appeared under different themes of research question.ie. Experiences of pregnant women, and partner’s involvement in decision making; it would therefore be affect themes if it were to be treated separately. For example partner’s role in buying dresses for pregnant women. Also this applies to what pregnant women and their friends have experienced while at the health facilities. Further discussion on this has been made as recommended. This is in the discussion section lines 590-596.

e. Safety surveillance: whilst this important, I’m not sure whether this is the first strategy that should be implemented. It’s also costly and will take time to implement. Consider other more-immediate interventions that may help these women. Some guidance in these papers:

i. Krishnaswamy S, Lambach P, Giles ML. Key considerations for successful implementation of maternal immunization programs in low and middle income countries. Human vaccines & immunotherapeutics. 2019 Apr 3;15(4):942-50.

ii. Ellingson MK, Dudley MZ, Limaye RJ, Salmon DA, O’Leary ST, Omer SB. Enhancing uptake of influenza maternal vaccine. Expert review of vaccines. 2019 Feb 1;18(2):191-204.

Response: The suggested articles have been helpful in identifying more interventions for recommendation that directly point to the findings of the study. These among others ongoing training health care professionals, incorporating vaccination into maternity care, strengthening antenatal care and surveillance systems. This is in the discussion section lines 649-667.

MINOR COMMENTS:

5. Title

a. The title should be revised. The word “determinants” is usually reserved for studies that have utilised quantitative methods.

Response: This has been revised to “Vaccines safety and maternal knowledge can boost maternal immunization acceptability in rural Uganda – A Qualitative Study Approach.” This is in the introduction section lines 1-2.

6. Article summary

a. Line 56: Is this about women in Uganda? And about maternal immunisation?

Response: This has been corrected to show reflect “…women in Uganda are generally knowledgeable about maternal immunization.” This is in the article summary section lines 62-63.

7. Key words

a. Suggest using more specific MeSH terms, such as “Health Knowledge, Attitudes, Practice” and a term to describe Uganda

Response: Thank you. Key words have been used as recommended and tested in the PubMed search engine to ensure that they are MeSH terms as recommended. This can be seen in introduction section, lines 49-51.

8. Introduction

a. Some of the sentences are quite long (e.g. lines 69-72). Consider shortening

Response: Thank you for pointing this out. The sentence has been shortened. This can be seen in introduction section, lines 77-80.

b. It’s important that the references used directly relate to the topic. For example, line 71 discusses tetanus, but reference #2 is not about tetanus\\

Response: Thank you for highlighting this. Reference 2 and 3 have been removed to replace relevant references that directly relate to what authors were communicating. This can be seen in introduction section, lines 80-81.

c. Line 86: How was 40% update for TT2+ estimated? What data was collected/used? Is it a reliable estimate?

Response: The statistic has been rectified to reflect the National coverage among pregnant women at only 49% in 2011. Data used by Ministry of Health (UNEPI program) is generated from District Health Information System (DHIS) that is periodically updated by district biostaticians as aggregated data. Quality may vary depending on who enters it and the completeness. We believe that it can at least provide an estimate reliable. This can be seen in introduction section, lines 94-96.

d. I suggest using systematic reviews as much as possible in the introduction. For example, the following systematic review should be referenced:

i. Nunes MC, Madhi SA. Influenza vaccination during pregnancy for prevention of influenza confirmed illness in the infants: a systematic review and meta-analysis. Human vaccines & immunotherapeutics. 2018 Mar 4;14(3):758-66.

Response: Systematic reviews have been replaced wherever possible that they exist and communicate the similar knowledge. This can be seen in introduction section, lines 80, 81, 88, 100, and 101.

9. Methods

a. When was the study carried out and how long did it go for? It would be worthwhile adding the year/s and duration into the ‘research method’ section

Response: A phrase to show the period and duration has been added as “for a period of 2 months between June 2019 and July 2019.” This can be seen in introduction section, line 162.

b. Participant recruitment and data collection:

i. As the reader, it would be clearer to move lines 157-162 to line 153, after “interviews at both the community and health facility.” This will help the reader understand how the research team purposively selected participants

Response: Thank you. As recommended, to maintain the flow of explanation of how purposive sampling was done, lines 157-162 have been moved to line 153.

10. Results

a. Table 1:

i. Given the numbers are <100, it is advisable to round the % to the nearest whole number

Response: Thank you. The percentages in the table have been rounded off to the nearest whole number as per recommendation. This can be seen in the results section lines 259-260.

ii. The total number in religion does not equal 90

Response: The error has been rectified to reflect 16 catholic women participating in the study. This can be seen in the results section line 259.

iii. A space is needed between the numbers and brackets

Response: This has been done for all numbers as recommended. This can be seen in the results section lines 259-260.

iv. Please spell out what TT means (underneath the table)

Response: The abbreviation for TT has been spelt out be underneath the table. This can be seen in the results section line 259.

b. Knowledge about maternal vaccines and immunisation:

i. Please clarify whether this sentence “Many were able to mention that vaccination prevents mothers and children from diseases like tetanus, measles, etc” is referring only to vaccination during pregnancy, or vaccination of people in general?

Response: This has been clarified to reflect vaccination during pregnancy. Note that this applied to both scenarios and it has been rectified to reflect so. This is reflected in results section lines 264-266.

c. Lines 357-358. I think the authors could expand on this and define what they mean by “sensitization.”

Response: Clarification has been provided on what sensitization they meant or required before taking up the new vaccine including; knowing the disease the vaccine prevents, schedule, when to receive it and its safety to both themselves and their unborn babies. This is reflected in results section lines 416-418.

d. Line 383: I suggest removing the Ugandan President’s name, and just refer to them as the President of Uganda. I.e., you would write “we are prepared but the problem is that they tell us that [The President of Uganda] wants to test the drugs on us….”

Response: This has been edited to remove Ugandan President’s name and maintain “[The President of Uganda]”. This can be seen in results section lines 446-447.

e. Lines 409-410: This quote doesn’t relate to the sentence above it. Consider using another quote.

Response: The section has been edited to make more sense for which the quote was intended. This can be seen in results section lines 475-478.

f. Line 421: Should “rugs” be “rags”?

Response: Thank you. This has been rectified. This can be seen in results section line 490.

11. Discussion

a. Lines 439-440: distinction is needed as to whether participants are talking about vaccination prior to pregnancy, or whether pregnant women think they haven’t yet been vaccinated against hepatitis B or HPV in their pregnancy?

Response: Thank you for pointing this out. A statement has been added to include"…. during their pregnancy while attending ANC.” This is because pregnant women were referring to vaccine received during the period when they confirm that they are pregnant and hence getting it at ANC. This can be seen in the discussion section lines 510-511.

b. Line 442: suggest using the word “prime” here instead of “encourage”

Response: Thank you. This has been used as per recommendation. This can be seen in the discussion section line 514.

c. Line 453: please reword “immune” to “vaccine.”

Response: Thank you. The word vaccine has be used instead of immune as advised. This can be seen in the discussion section line 528.

Reviewer #2 Point by Point Response to Review of manuscript for PLOS ONE

Summary: This is a well done manuscript. The title, background, findings, and discussion are very well laid out. Methods need some clarity for reproducibility. Just a few suggestions to make it a little more crisp:

Comments on

Title: No comment

Abstract: 

Lines 33 & 34: You need to recast this sentence. “Women expressed willingness to take new vaccines, positive attitudes towards maternal immunization, and were familiar with its importance”. Recast the sentence to follow the order in your objectives namely knowledge, attitudes and willingness. For example: “Women were familiar with the importance of maternal vaccines, have positive attitudes and expressed willingness to take them”.

Response: Thank you. The statement has been edited as advised to reflect a better flow and coherent of events being discussed. This can be seen in abstract section on lines 32-35.

Line 35: “……... affected by adverse events”, ie changing the effects to events. Also recast the statement to be more sharp and crisp. Such as “………. affected by worries of pregnant women and that of their partners’ who influence health seeking decisions in a home concerning adverse events following the maternal immunization.”

Response: Thank you. The statement has been revised as recommended to reproduce the recommendation. This can be seen in abstract section on lines 35-37.

Line 36-38: “Misconceptions about introduction of vaccines like thinking that vaccines will be tested on only them before introduction in larger population, and that vaccines treat illnesses like malaria and general body weakness.” Consider this phrase “ …… of vaccines such as the belief that vaccines treat malaria and general body weakness and that of being used as guinea-pig to test for the vaccine before its introduction to the larger population.” 

Response: Thank you. This line has been rephrased as per recommendation of the reviewer. This can be seen in abstract section on lines 39-42.

Background:

Line 77: It is World Health Organization and not “world ….” 

Response: Thank you. This has been corrected. This can be seen in introduction section on line 85.

Line 80: “………. are administered to pregnant women.”

Response: Thank you. This has also been corrected. This can be seen in introduction section on line 90.

Line 82: Do you mean that poor neonatal health outcomes are due to lack of studies on knowledge ………. Make it clearer.

Response: Thank you. The sentence was separated and made clearer to bring out the fact that while evidence is limited, poor neonatal health outcomes continue to happen. This can be seen in introduction section on lines 92-94.

Line 87: Be consistent. “ Immunisation” is British English. You have been using American English. In addition, there is a little flaw in the content and mechanical accuracy in that sentence. You could consider “This can partly be attributed to limited knowledge among pregnant women, their poor attitude about immunization, their failure to attend all ANC visits, lack of training of the Village Health Teams (VHTs) on the importance of TT vaccination for pregnant mothers and limited health education to pregnant mothers.” 

Response: Thank you. Immunisation had been corrected to Immunization. See line 97. The mistake has been rectified using the recommendation. See lines 96-100.

Line 92: “ ………. middle-income countries in respect of Pneumococcal Conjugate Vaccine ……….. “

Response: Thank you. This has been used to rectify the mistake highlighted in line 92. This can be seen on line 109.

Line 95: GAVI means Global Alliance for Vaccines and Immunizations.

Response: Thank you. Mistake on how GAVI was written has been corrected using the recommendation. This can be seen on line 111.

Methods: This section need some work. 

Line 144: You had included health workers in the research population included. You failed to include them now.

Response: These have also now been included in the research population as earlier stated and in the recommendation. This can be seen on lines 169-170.

Line 153: Which of the population had FGD and which one had KII. You need to be coherent and orderly.

Response: Thank you. Clarification has been provided that FGD were done among pregnant women and KII were done among health care workers. Statement have been arranged to ensure coherence as well. This can be seen on lines 176-177.

Line 157: Readers may find it difficult to understand what you mean. I think you may look at it again and recast it to bring out your points clearly.

Response: Thank you. Line 157 has been edited to ensure clarity in communication. Statement was too long to follow. It has been split into two and can easily be understood now. It says “…….all pregnant women available at ANC and maternity points of care in June and July 2019 at the visited health facilities in Iganga. During the same period, pregnant women in the communities were identified and invited by the VHTs for FGDs. This can be seen on lines 177-183.

Line 165: Order and sequence is important as mentioned in previous comment. 

Response: Thank you. This has been rectified to ensure order and coherence when one is reading. This can be seen on lines 190-191.

Line 174: Pointless mentioning names of those that played a role here.

Response: Thank you. We felt that for transparency reasons, the names and role played be left in the manuscript as we have observed in other similar papers. This can be seen on lines 192-193.

Results:

Line 194: “(2 FGDs pregnant women and 2 health workers) could read “(2 FGDs for pregnant women and 2 KIIs for 2 health workers)”, or what exactly, do you have in mind.

Response: Thank you. Statement was rectified to read as “….(2 FGDs for pregnant women and 2 KIIs for 2 health workers) to evaluate the questions for their cultural appropriateness to the target population.” This can be seen on lines 234-235.

Line 208: Change the FDG to FGD. Where exactly is the study area; rural or urban or rural and semi-urban? Be consistent.

Response: Thank you. FDG has been changed to FGD. A study area is in rural and semi-urban and this has been made consistent throughout. This can be seen on lines 248 and 252.

Line 239: You use either “reason” or “why”.

Response: Thank you. The word “reason” has been removed to remain with “why”. This can be seen on line 282.

Line 250: I could have preferred “….one woman added” than the word chipped in. This is scientific writing.

Response: Thank you. The phrase “chipped in” has been replaced with “one woman added” as recommended. This can be seen on line 293.

Line 264: AEFI means adverse events following immunization. It refers to any untoward medical occurrence which follows immunization and which does not necessarily have a causal relationship with the usage of the vaccine.

Response: Thank you. This has been noted and consistency of using AEFI has been considered. This can be seen on lines 307-308.

Line 283: could read “ …… highlighted that some of ……”

Response: Thank you for the suggestion. This has also been rectified using the recommendation. This can be seen on line 326.

Line 290: “Participants also expressed that some pregnant women did not go for vaccination due to their prohibitive religious practice like some Pentecostal churches”, could read “Participants …….. due to their religious belief that prohibites vaccination”.

Response: Thank you. This has been rectified using the recommendation. This can be seen on lines 333-335.

Line 296: This statement “Some mentioned the fear for injections and HIV testing (which is compulsory for all women attending ANC) would be the reason why some women do not seek maternal vaccination” could read “Fear of injections and HIV testing could be the reason some women fail to seek maternal vaccination.” 

Response: Thank you. This has been rectified using the recommendation. This can be seen on lines 340-341.

Line 369: “When health workers were asked what women are worried about, only one mentioned AE worries from the pregnant women and their partners” could read “When health workers were asked what women are worried about, only one mentioned that the pregnant women and their partners were concerned about AEFI.”

Response: Thank you. This has been rectified using the recommendation. This can be seen on lines 429-430.

Line 375: The statement could be rephrased thus “Some participants believe that they were used to test for the safety of new vaccines and thus they would not take the vaccines until its safety is confirmed by others who had received it.

Response: Thank you. This has been rectified using the recommendation. This can be seen on lines 435-438.

Line 389: Could be rephrased as “Pregnant women …………. vaccination even without accompanying them and their children to immunization centers.”

Response: Thank you. This has been rectified using the recommendation. This can be seen on lines 452-454.

Line 400: You may want to rephrase thus “Health workers cited that they encouraged pregnant women to come with their partners for ANC visits.”

Response: Thank you. This has been rectified using the recommendation. This can be seen on lines 464-465.

Line 417: “…………. not effective since women are not comfortable with moving with husbands and more so their husbands don’t dress well.”

Response: Thank you. This has been rectified using the recommendation. This can be seen on lines 484-487.

Discussion: 

Line 434: be consistent with the study area.

Response: Thank you. The study area has been made the same as earlier suggested. This can be seen on line 503.

Line 452: change adverse effect to adverse event

Response: Thank you. Adverse effect has been changed to adverse event. This can be seen on lines 527-528.

Line 475: Rephrase as “…………... and religious practices found in some Pentecostal churches, that prohibite maternal vaccination.

Response: Thank you. This has been rectified using the recommendation. This can be seen on lines 551-552.

Line 482: Rephrase the statement for clarity

Response: Thank you. This has been rectified using the earlier recommendation on AEFI and also more sensitization has been added. This can be seen on line 558.

Line 486: Rephrase the statement. It could read “Some believe that the new vaccines …………...”

Response: Thank you. This has been rectified using the recommendation. This can be seen on lines 562-563.

Line 502: delete treatment from the sentence.

Response: Thank you. The word “treatment” has been deleted from the sentence. This can be seen on line 562.

Line 504: You can rephrase as “The attitudes and behaviors of health care providers towards pregnant women is an …….”

Response: Thank you. This has been rephrased using the recommendation. This can be seen on line 579.

Line 517: Consider rephrasing thus, “……… get vaccinated for their good and that of their children”.

Response: Thank you. This has been rephrased using the recommendation. This can be seen on line 583-584.

Line 518: Consider this rephrasing “The partner’s worries about AEFI was an important concern for women ………….”

Response: Thank you. This has been rephrased using the recommendation.

Line 537: See the comment on line 34

Response: Thank you. Adjustment have been made to ensure order and coherence while writing as per the research question. This can be seen on line 610.

Line 538: Define ADR. Do you mean Adverse Drug Reaction? Since we are talking of immunization why not use AEFI instead.

Response: Thank you. This has been corrected to AEFI. This can be seen on line 618.

---

## [Decision Letter · Decision Letter 1]

30 Oct 2020

PONE-D-20-15557R1

Vaccines safety and maternal knowledge for enhanced maternal immunization acceptability in rural Uganda – A Qualitative Study Approach.

PLOS ONE

Dear Dr. Kajungu,

Thank you for submitting your manuscript to PLOS ONE. After careful consideration, we feel that it has merit but does not fully meet PLOS ONE’s publication criteria as it currently stands. Therefore, we invite you to submit a revised version of the manuscript that addresses the points raised during the review process.

We look forward to receiving your revised manuscript.

Kind regards,

Holly Seale

Academic Editor

PLOS ONE

Reviewers' comments:

Reviewer's Responses to Questions

**Comments to the Author**

1. If the authors have adequately addressed your comments raised in a previous round of review and you feel that this manuscript is now acceptable for publication, you may indicate that here to bypass the “Comments to the Author” section, enter your conflict of interest statement in the “Confidential to Editor” section, and submit your "Accept" recommendation.

Reviewer #2: All comments have been addressed

2. Is the manuscript technically sound, and do the data support the conclusions?

Reviewer #2: Yes

3. Has the statistical analysis been performed appropriately and rigorously? 

Reviewer #2: Yes

4. Have the authors made all data underlying the findings in their manuscript fully available?

Reviewer #2: Yes

5. Is the manuscript presented in an intelligible fashion and written in standard English?

Reviewer #2: No

6. Review Comments to the Author

Reviewer #2: Good job. You just need to re-read the article painstakingly for few grammatical errors, most of which I have pointed out. See the attachment.

7. PLOS authors have the option to publish the peer review history of their article (what does this mean?). If published, this will include your full peer review and any attached files.

Reviewer #2: **Yes: **Dr Uchechukwu Joel Okenwa

---

## [Author Response · Author response to Decision Letter 1]

18 Nov 2020

Reviewer #2: Good job. You just need to re-read the article painstakingly for few grammatical errors, most of which I have pointed out. See the attachment

Response: Thank you very much for the guidance and pointing out the grammatical errors. We have reviewed the manuscript, corrected the errors, clarified where necessary and improved the flow of the manuscript.

Editor's comment:

1) Please ensure that you refer to Table 2 in your text as, if accepted, production will need this reference to link the reader to the Table.

Response: Table2 was added to the manuscript by mistake. The information is referenced as SI File 3 in the results section

---

## [Editor Report · Decision Letter 2]

27 Nov 2020

Vaccines safety and maternal knowledge for enhanced maternal immunization acceptability in rural Uganda – A Qualitative Study Approach.

PONE-D-20-15557R2

Dear Dr. Kajungu,

We’re pleased to inform you that your manuscript has been judged scientifically suitable for publication and will be formally accepted for publication once it meets all outstanding technical requirements.

Kind regards,

Holly Seale

Academic Editor

PLOS ONE
---

## [Editor Report · Acceptance letter]

2 Dec 2020

PONE-D-20-15557R2 

Vaccines safety and maternal knowledge for enhanced maternal immunization acceptability in rural Uganda – A Qualitative Study Approach. 

Dear Dr. Kajungu:

I'm pleased to inform you that your manuscript has been deemed suitable for publication in PLOS ONE. Congratulations! Your manuscript is now with our production department. 

Kind regards, 

on behalf of

Dr. Holly Seale 

Academic Editor

PLOS ONE